# Cold Brew: Distilling Graph Node Representations with Incomplete or Missing Neighborhoods

**Wenqing Zheng, Edward W Huang, Nikhil Rao, Sumeet Katariya, Zhangyang Wang, Karthik Subbian**

{`wenqzhen,ewhuang,nikhilsr,katsumee,wzhangwa,ksubbian`}@amazon.com

## Abstract

Graph Neural Networks (GNNs) have achieved state-of-the-art performance in node classification, regression, and recommendation tasks. GNNs work well when rich and high-quality connections are available. However, their effectiveness is often jeopardized in many real-world graphs in which node degrees have power-law distributions. The extreme case of this situation, where a node may have no neighbors, is called Strict Cold Start (SCS). SCS forces the prediction to rely completely on the node's own features. We propose **Cold Brew**, a teacher-student distillation approach to address the SCS and noisy-neighbor challenges for GNNs. We also introduce feature contribution ratio (FCR), a metric to quantify the behavior of inductive GNNs to solve SCS. We experimentally show that FCR disentangles the contributions of different graph data components and helps select the best architecture for SCS generalization. We further demonstrate the superior performance of Cold Brew on several public benchmark and proprietary e-commerce datasets, where many nodes have either very few or noisy connections. Our source code is available at `https://github.com/amazon-research/gnn-tail-generalization`.

## 1 Introduction

Graph Neural Networks (GNNs) achieve state-of-the-art results across a wide range of tasks such as graph classification, node classification, link prediction, and recommendation (Wu et al., 2020; Goyal & Ferrara, 2018; Kherad & Bidgoly, 2020; Shaikh et al., 2017; Silva et al., 2010; Zhang et al., 2019). Most modern GNNs rely on the principle of message passing to aggregate each node's features from its (multi-hop) neighborhood (Kipf & Welling, 2016; Veličković et al., 2017; Hamilton et al., 2017; Xu et al., 2018a; Wu et al., 2019; Klicpera et al., 2018). Therefore, the success of GNNs relies on the presence of dense and high-quality connections. Even inductive GNNs Hamilton et al. (2017) learn a function of the node feature and the node neighborhood, which requires the neighborhood to be present during inference.

A practical barrier for widespread applicability of GNNs arises from the long-tail node-degree distribution existing in many large-scale real-world graphs. Specifically, the node degree distribution is power law in nature, with a majority of nodes having very few connections (Hao et al., 2021; Ding et al., 2021; Lam et al., 2008; Lu et al., 2020). Figure 1 (top) illustrates a long-tail distribution, accompanied with the statistics of several public datasets (bottom).

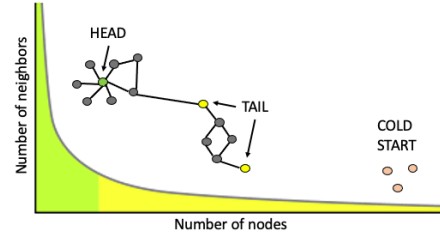
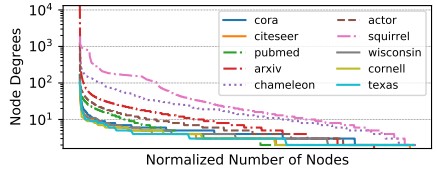

Figure 1: **Top:** Graph nodes may have a power-law ("long-tail") connectivity distribution, with a large fraction of nodes (yellow) having few to no neighbors. **Bottom:** Long-tail distributions in real-world datasets, making modern GNNs fail to generalize to the tail/cold-start nodes.

Many information retrieval and recommendation applications face the scenario of *Strict Cold Start* (**SCS**) (Li et al., 2019b; Ding et al., 2021), wherein some nodes have no edges connected. Predicting for these nodes admittedly is even more challenging than the tail nodes in the graph. In these cases, existing GNNs fail to perform well due to the sparsity or absence of the neighborhood.

In this paper, we develop GNN models that have *truly inductive* capabilities: one can learn effective node embeddings for "orphaned" nodes in a graph. This capability is important to fully realize the potential of large-scale GNN models on modern, industry-scale datasets with very long tails and many orphaned nodes. To this end, we adopt the teacher-student knowledge distillation procedure (Yang et al., 2021; Chen et al., 2020b) and propose **Cold Brew** to distill the knowledge of a GNN teacher into a multilayer perceptron (MLP) student.

The Cold Brew framework addresses two key questions: (1) how we can efficiently distill the teacher's knowledge for the sake of tail and cold-start generalization, and (2) how can a student make use of this knowledge. We answer these two questions by learning a latent node-wise embedding using knowledge distillation, which both avoids "over-smoothness" (Oono & Suzuki, 2020; Li et al., 2018; NT & Maehara, 2019) and discovers latent neighborhoods, which are missing for the SCS nodes. Note that in contrast to traditional knowledge distillation (Hinton et al., 2015), our aim is not to train a simpler student model to perform as well as the more complex teacher. Instead, we aim to train a student model that is better than the teacher in terms of generalizing to tail or SCS samples.

In addition, to help select the cold-start friendly model architectures, we develop a metric called *Feature Contribution Ratio* (FCR) that quantifies the contribution of node features with respect to the adjacency structure in the dataset for a specific downstream task. FCR indicates the difficulty level in generalizing to tail and cold-start nodes and guides our principled selection of both teacher and student model architectures in Cold Brew. We summarize our key contributions as follows:

- To generalize better to tail and SCS nodes, we design the Cold Brew knowledge distillation framework: we enhance the teacher GNN by appending the node-wise Structural Embedding (SE) to strengthen the teacher's expressiveness, and design a novel mechanism for the MLP student to rediscover the missing "latent/virtual neighborhoods," on which it can perform message passing.
- We propose Feature Contribution Ratio (FCR), which quantifies the difficulty in generalizing to tail and cold-start nodes. We leverage FCR in a principled "screening process" to select the best model architectures for both the GNN teacher and the MLP student.
- As the existing GNN studies only evaluate on the entire graph and do not explicitly evaluate on head/tail/SCS, we uncover the hidden differences of head/tail/SCS by creating bespoke train/test splits. Extensive experiments on public and proprietary e-commerce graph datasets validate the effectiveness of Cold Brew in tail and cold-start generalization.

## 1.1 PROBLEM SETUP

GNNs effectively learn node representations using two components in graph data: they process *node features* through distributed node-wise transformations and process *adjacency structure* through localized neighborhood aggregations. For the first component, GNNs apply shared feature transformations to all nodes regardless of the neighborhoods. For the second component, GNNs use permutation-invariant aggregators to collect neighborhood information.

We take the node classification problem in the sequel for the sake of simplicity. All our proposed methods can be easily adapted to other semi-supervised or unsupervised problem settings, which we show in Section 5. We denote the graph data of interest by $\mathcal{G}$ with node set $\mathcal{V}$, $|\mathcal{V}| = N$. Each node possesses a $d_{in}-$dimensional feature and a $d_{out}-$dimensional label (either $d_{out}$ classes or a continuous vector in the case of regression). Let $\mathbf{X}^0 \in \mathbb{R}^{N \times d_{in}}$ and $\mathbf{Y} \in \mathbb{R}^{N \times d_{out}}$ be the matrices of node features and labels, respectively. Let $\mathcal{N}_i$ be the neighborhood of the $i$-th node, $0 \leq i < N$. In large-scale graphs, $|\mathcal{N}_i|$ is often small for a (possibly substantial) portion of nodes. We refer to these nodes as *tail nodes*. Some nodes may have $|\mathcal{N}_i| = 0$, and we refer to these extreme cold start cases as *isolated nodes*.

A classical GNN learns representations for the $i^{th}$ node at the $l^{th}$ layer as a function of its representation and its neighborhood's representations at the $(l-1)^{th}$ layer:

$$x_i^l := f\left(\{x_i^{l-1}\}, \{x_j^{l-1}\}_{j \in \mathcal{N}_i}\right) \tag{1}$$

where $f(\cdot)$ is a general function that applies node-wise transformation on node $x_i^{l-1}$ and aggregates information of its neighborhood $\{x_j^{l-1}\}_{j \in \mathcal{N}_i}$ to obtain the final node representation. Given $i$'s input features $x_i^0$ and its neighborhood $\mathcal{N}_i$, one can use (1) to obtain its representation and predict $y_i$, making these models inductive.

We are interested in improving the performance of these GNNs on a set of tail and cold-start nodes, where $\mathcal{N}_i$ for node $i$ is either unreliable[1] or absent. In these cases, applying (1) will yield a suboptimal node representation, since $\{x_j^{l-1}\}_{j \in \mathcal{N}_i}$ will be unreliable or empty at inference time.

## 2 RELATED WORK

GNNs learn by aggregating neighborhood information to learn node representations (Kipf & Welling, 2016; Veličković et al., 2017; Hamilton et al., 2017; Xu et al., 2018a; Wu et al., 2019; Klicpera et al., 2018). Inductive variants of GNNs such as GraphSAGE (Hamilton et al., 2017) require initial node features as well as the neighborhood information of each node to learn the representation. Most works on improving GNNs have focused on learning better aggregation functions, and methods that can work when the neighborhood is absent or noisy have not been sufficiently exploited, except two recent concurrent works (Hu et al., 2021; Zhang et al., 2021).

In the context of cold start, (Hao et al., 2021) and (Ding et al., 2021) employ a transfer learning approach. (Yang et al., 2021) proposes a knowledge distillation approach for GNN, while (Chen et al., 2020b) proposes a self-distillation approach. In all the above cases, the models need full knowledge of the neighbors of the cold-start nodes in question and do not address the case of noisy or missing neighborhoods. Another possible solution is to directly train an MLP that only takes node features. (Hu et al., 2021) proposes to learn graph embeddings with only node-wise MLP, while using contrastive loss to regularize the graph structure.

Some previous works have studied the relation between node feature similarity and edge connections and how that influences the selection of appropriate graph models. (Pei et al., 2020) proposed the homophily metric that categorizes graphs into assortative and disassortative classes. (Wang et al., 2021) dissected the feature propagation steps of linear GCNs from a perspective of continuous graph diffusion and analyzed why linear GCNs fail to benefit from more propagation steps. (Liu et al., 2020a) further studied the influence of homophily on model selection and proposed a non-local GNN.

## 3 STRICT COLD START GENERALIZATION

We now address the problem of generalization to the tail and cold-start nodes, where the neighborhood information is missing/noisy (Section 1). A naive baseline is to train an MLP to map node features to labels. However, such a method would disregard all graph information, and we show via our Feature Contribution Ratio and other experimental results that for most assortative graph datasets, the node-wise MLP approach is suboptimal.

The key idea of our framework is the following: the GNN maps node features into a $d$-dimensional embedding space, and since the number of nodes $N$ is usually much bigger than the embedding dimensionality $d$, we end up with an overcomplete set for this space using the embeddings as the basis. This implies the possibility that any node representation can be cast as a linear combination of $K \ll N$ existing node representations. Our aim will be to train a student model that can accurately discover the combination of the best $K$ existing node embeddings of a target isolated node. We call this procedure *latent/virtual neighborhood discovery*, which is equivalent to using MLPs to "mimic" the node representations learned by the teacher GNN.

We adopt the knowledge distillation procedure (Yang et al., 2021; Chen et al., 2020b) to improve the quality of the learned embeddings for tail and cold-start nodes. We use a teacher GNN model to embed the nodes onto a low-dimensional manifold by utilizing the graph structure. Then, the goal of the student is to learn a mapping from the node features to this manifold without knowledge of the graph that the teacher has. We further aim to let the student model generalize to SCS cases where the teacher model fails, beyond just mimicking the teacher as standard knowledge distillation does.

---

[1]For example, a user with only one movie watched or an item with too few purchases.

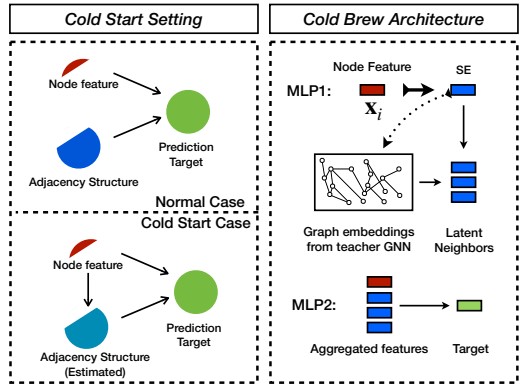
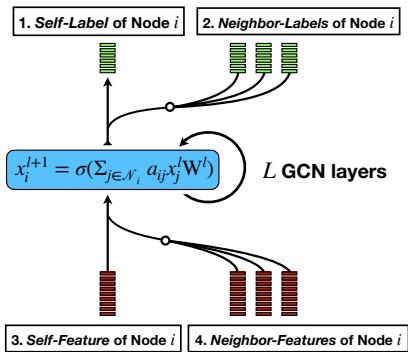

(a) The teacher-student knowledge distillation of the Cold Brew framework under the cold-start setting.

(b) Four GNN atomic components in deciding GNN's output, which are used for FCR analysis.

Figure 2: **(a)**: The proposed Cold Brew framework. In normal case (left upper), GNN relies on both node feature and adjacency structure to make prediction. In cold start case (left lower) when the adjacency structure is missing, the cold brew student model first estimate the adjacency structure, then use both node feature and adjacency structure to make prediction. The "SE" (right) is the structural embedding learned by Cold Brew's teacher GNN. **(b)**: Four atomic components deciding the GNN embeddings of node $i$. Our proposed FCR metric disentangles them into two models: the MLP that only considers Part 1 and Part 3, and label propagation that only considers Part 1 and Part 2.

### 3.1 THE TEACHER MODEL OF COLD BREW: STRUCTURAL EMBEDDING GNN

Consider a graph $\mathcal{G}$. For a Graph Convolutional Network with $L$ layers, the $l$-th layer transformation can be written as[2]: $\mathbf{X}^{(l+1)} = \sigma(\tilde{\mathbf{A}}\mathbf{X}^{(l)}\mathbf{W}^{(l)})$, where $\tilde{\mathbf{A}}$ is the normalized adjacency matrix, $\tilde{\mathbf{A}} = \mathbf{D}^{-1/2}\mathbf{A}\mathbf{D}^{-1/2}$, $\mathbf{D}$ is the diagonal degree matrix, and $\mathbf{A}$ is the adjacency matrix. $\mathbf{X}^{(l)} \in \mathbb{R}^{N \times d_1}$ is the node representations in the $l$-th layer, $\mathbf{W}^{(l)} \in \mathbb{R}^{d_1 \times d_2}$ is the feature transformation matrix, where the values of $d_1/d_2$ depend on layer $l$: $(d_1, d_2) = (d_{in}, d_{hidden})$ for $l = 0$, $(d_{hidden}, d_{hidden})$ for $1 \leq l \leq L - 2$, and $(d_{hidden}, n_{classes})$ for $l = L - 1$. $\sigma(\cdot) =$ is the nonlinear functions applied to each layer, (e.g., ReLU). $Norm(\cdot)$ refers to an optional batch or layer normalization.

GNNs typically suffer from oversmoothing (Oono & Suzuki, 2020; Li et al., 2018; NT & Maehara, 2019), i.e., node representations become too similar to each other. Inspired by the positional encoding in Transformers (Vaswani et al., 2017), we train the teacher GNN to learn an additional set of node embeddings that can be appended, which we term the **Structural Embedding (SE)**. SE learns to incorporate extra information besides original node features (such as node labels in the case of semi-supervised learning) through gradient backpropagation. The existence of SE avoids the oversmoothing issue in GNNs: the transformations applied to different nodes are no longer the same for every node since the SE of each node is different and participates in the feature transformation. This can be of independent interest to GNN researchers.

Specifically, for each layer $l, 0 \leq l \leq L - 1$, the Structural Embedding takes the form of a learnable matrix $\mathbf{E}^{(l)}$, and the SE-GNN layer forward pass can be written as:

$$\mathbf{X}^{(l+1)} = \sigma\left(\tilde{\mathbf{A}}\left(\mathbf{X}^{(l)}\mathbf{W}^{(l)} + \mathbf{E}^{(l)}\right)\right), \mathbf{X}^{(l)} \in \mathbb{R}^{N \times d_1}, \mathbf{W}^{(l)} \in \mathbb{R}^{d_1 \times d_2}, \mathbf{E}^{(l)} \in \mathbb{R}^{N \times d_2} \quad (2)$$

**Remark 1:** Note that SE is not the same as the bias term in traditional feature transformation $\mathbf{X}^{(l+1)} = \sigma\left(\tilde{\mathbf{A}}\left(\mathbf{X}^{(l)}\mathbf{W}^{(l)} + \mathbf{b}^{(l)}\right)\right)$; in the bias $\mathbf{b} \in \mathbb{R}^{N \times d_2}$, the rows are copied/shared across all nodes. In contrast, we have a different structural embedding for every node.

**Remark 2:** SE is also unlike traditional label propagation (LP) (Iscen et al., 2019; Wang & Leskovec, 2020; Huang et al., 2020b). LP encodes label information through iterating $\mathbf{E}^{(t+1)} = (1 - \alpha)\mathbf{G} + \alpha\tilde{\mathbf{A}}\mathbf{E}^{(t)}$, where $\mathbf{G}$ is a one-hot encoding of ground truth for training node classes and zeros for test nodes, and $0 < \alpha < 1$ is the portion of mixture at each iteration.

---

[2]Compared to Equation (1), multiplication by $\tilde{\mathbf{A}}$ plays the role of aggregating both $\{x_i\}$ and $\{x_j\}_{i \in \mathcal{N}_i}$.

SE-GNN enables node $i$ to learn to encode the self and neighbors' label information[3] into its own node embedding through $\tilde{A}$. We use the Graph Convolutional Networks (Kipf & Welling, 2016), combined with other building blocks proposed in recent literature including: (1) initial/dense/jumping connections, and (2) batch/pair/node/group normalization as the backbone of Cold Brew's teacher GNN. More details are described in Appendix B. We also apply a regularization term to the loss function, yielding the following loss function:

$$loss = CE(\mathbf{X}_{train}^{(L)}, \mathbf{Y}_{train}) + \eta \|\mathbf{E}\|_2^2 \tag{3}$$

where $\mathbf{X}_{train}^{(L)}$ is the model's embedding at the $L$-th layer, $CE(\mathbf{X}_{train}^{(L)}, \mathbf{Y}_{train})$ is the Cross Entropy loss between the model output $\mathbf{X}_{train}^{(L)}$ and the ground truth $\mathbf{Y}$ on the training set, and $\eta$ is a regularization coefficient (grid-searched for different datasets in practice). The Cross-Entropy loss can be replaced by any other appropriate loss depending on the task.

## 3.2 THE STUDENT MLP MODEL OF COLD BREW

We design the student to be composed of two MLP modules. Given a target node, the first MLP module imitates the node embeddings generated by the GNN teacher. Next, given any node, we find a set of virtual neighbors of that node from the graph. Finally, a second MLP attends to both the target node and the virtual neighborhood and transforms them into the embeddings of interest.

Suppose we would like to obtain the embedding of a potentially isolated target node $i$ given only its feature $\mathbf{x}_i$. From the teacher GNN, at each layer $l$, we have access to two sets of node embeddings: $\mathbf{X}^{(l)}\mathbf{W}^{(l)}$ and $\mathbf{E}^{(l)}$. Denote $\bar{\mathbf{E}}$ as the embeddings that the teacher GNN passes over to the student MLPs. We offer two options for $\bar{\mathbf{E}}$: it can be the final output of the teacher GNN (in this case, $\bar{\mathbf{E}} \in \mathbb{R}^{N \times d_{out}} := \mathbf{X}^{(L)}$), or it can be the concatenation of all intermediate results of the teacher GNN, similar to (Romero et al., 2014): $\bar{\mathbf{E}} \in \mathbb{R}^{N \times (d_{hidden}*(L-1)+d_{out})} := \mathbf{X}^{(L)} \bigcup_{l=0}^{L-1}(\mathbf{E}^{(l)} + \mathbf{X}^{(l)}\mathbf{W}^{(l)})$, where $\bigcup$ is the concatenation of matrices at the feature dimension (second dimension). $\bar{\mathbf{E}}$ acts as the target for the first MLP and also the input to the second MLP.

The first MLP learns a mapping from the input node features $\mathbf{X}^{(0)}$ to $\bar{\mathbf{E}}$, i.e., for node $i$, $\hat{e}_i = \xi_1(\mathbf{x}_i^{(0)})$, where $\hat{e}_i$ is trained with supervised learning to reproduce $\bar{\mathbf{E}}[i, :]$. Then, we discover the virtual neighborhood by applying an attention-based aggregation of the existing embeddings in the graph before linearly combining them:

$$\tilde{e}_i = softmax(\Theta_K(\hat{e}_i\bar{\mathbf{E}}^\top))\bar{\mathbf{E}} \tag{4}$$

where $\Theta_K(\cdot)$ is the top-$K$ hard thresholding operator: for $z \in \mathbb{R}^{1 \times N}$: $[\Theta_K(z)]_j = z_j$ if $z_j$ is among the top-$K$ largest elements of $z$, and $\Theta_K(z)_j = -\infty$ otherwise. Finally, the second MLP learns a mapping $\xi_2 : [\mathbf{x}_i, \tilde{e}_i] \to \mathbf{y}_i$, where $\mathbf{y}_i = \mathbf{Y}[i, :]$ is the ground truth for node $i$.

Equation (4) first selects $K$ nodes from the $N$ nodes that the teacher GNN was trained on via the hard thresholding operator. $\tilde{e}_j$ is then a linear combination of $K$ node[4] embeddings. Thus, every sample whether or not seen previously while training the GNN can be represented as a linear combination of these representations. The MLP $\xi_2(\cdot)$ maps this representation to the final target of interest. Thus, we decompose every node embedding as a linear combination of an (overcomplete) basis.

The training of $\xi_1(\cdot)$ occurs by minimizing the mean squared error over the non-isolated nodes in the graph (mimicking the teacher's embeddings), and the training of $\xi_2(\cdot)$ occurs by minimizing the cross entropy (for the node classification task) or mean squared error (for the node regression task) on the training split of the tail and isolated part of the graph. An illustration of SE-MLP's inference procedure for the isolated nodes is shown in Figure 2. When the number of nodes is large, the ranking procedure involved in $\Theta_K(\cdot)$ can be precomputed after training the first part and before training the second part.

## 3.3 MODEL INTERPRETATION FROM A LABEL SMOOTHING PERSPECTIVE

We quote Theorem 1 in (Wang & Leskovec, 2020): *Suppose that the latent ground-truth mapping from node features to node labels is differentiable and L-Lipschitz. If the edge weights $a_{ij}$ approximately*

---

[3]This will be inferred in the case of missing labels.

[4]We abuse terminology here since $\mathbf{E}$ contains node and structural embeddings from multiple layers.

*smooth $\mathbf{x}_i$ over its immediate neighbors with error $\epsilon_i$, i.e., $\mathbf{x}_i = \frac{1}{d_{ii}}\Sigma_{j \in \mathcal{N}} a_{ij}\mathbf{x}_j + \epsilon_i$, then the $a_{ij}$ also approximately smooth $y_i$ to bound within error $|y_i - \frac{1}{d_{ii}}\Sigma_{j \in \mathcal{N}_i} a_{ij}y_j| \leq L||\epsilon||_2 + o(\max_{j \in \mathcal{N}_i}(||\mathbf{x}_j - \mathbf{x}_i||_2))$, where $o(\cdot)$ denotes a higher order infinitesimal.*

This theorem indicates that the errors of the label predictions are determined by the difference of the features after neighborhood aggregation: if $\epsilon_i$ is large, then the error in the label prediction is also large, and vice versa. However, with structural embeddings, each node $i$ also learns an independent embedding $\bar{\mathbf{E}}[:, i]$ during the aggregation, which changes $\frac{1}{d_{ii}}\Sigma_{j \in \mathcal{N}} a_{ij}\mathbf{x}_j + \epsilon_i$ into $\frac{1}{d_{ii}}\Sigma_{j \in \mathcal{N}} a_{ij}\mathbf{x}_j + \bar{\mathbf{E}}[:, i] + \epsilon_i$. Deduced from this theorem, the structural embedding $\bar{\mathbf{E}}$ is important for the teacher model: it allows higher flexibility and expressiveness in learning the residual difference between nodes, and hence the error $\epsilon_i$ can be lowered if $\bar{\mathbf{E}}$ is properly learned.

From this theorem, one can also see the necessity of introducing neighborhood aggregations like that of the Cold Brew student model. If one directly applies MLP models without neighborhood aggregation, the $\epsilon_i$ turns out to be non-negligible, leading to higher losses in the label predictions. However, Cold Brew introduces the neighborhood aggregation mechanism so that the second part of the student MLP takes over the aggregation of neighborhood generated by the first MLP. Therefore, Cold Brew eliminates the above residual error even in the absence of the actual neighborhood.

## 4 MODEL SELECTION AND GRAPH COMPONENT DISENTANGLEMENT WITH FEATURE CONTRIBUTION RATIO

We now discuss Feature Contribution Ratio (FCR), a metric to quantify the difficulty of learning representations under the truly inductive cold-start case, and a hyperparameter optimization approach to select the best suitable model architecture that helps tail and cold-start generalization.

As conceptually illustrated in Figure 2, there are four atomic components contributing to the learned embedding of node $i$ in the graph: 1. the label of $i$ (*self-label*); 2. the label of neighbors of $i$ (*neighbor-labels*); 3. the features of $i$ (*self-feature*); 4. the features of neighbors of $i$ (*neighbor-features*). To quantize the SCS generalization difficulty, we first divide these four components into two submodules to disentangle the contributions of the *node features* with respect to the *adjacency structure* of the graph dataset. Then, we quantize it based on the assumption that the SCS generalization difficulty is proportional to the contribution ratio of the *node features*.

We posit that a submodule that learns accurate node representations must include the node's (self) label, so that training can be performed via backpropagation. What remains is to use the label with other atomic components to construct two specialized models that each make use of only the node features or the adjacency structure. For the first submodule, we build an MLP that maps the self-features to self-labels, ignoring any neighborhood information present in the dataset. For the second submodule, we adopt a Label Propagation (LP) method (Huang et al., 2020a)[5] to learn representations from self- and neighbor-labels. This model ignores any node feature information.

With the above two submodules, we introduce the Feature Contribution Ratio (FCR) that characterizes the relative importance of the node features and the graph structure. Specifically, for graph dataset $\mathcal{G}$, we define the contribution of a submodule to be the *residual performance* of the submodule compared to a full-fledged GNN (e.g., Equation (1)) using both the node feature as well as the adjacency structure. Denote $z_{MLP}, z_{LP}$, and $z_{GNN}$ as the performance of the MLP submodule, LP submodule, and the full GNN on the test set, respectively. If $z_{MLP} \ll z_{GNN}$, then $FCR(\mathcal{G})$ is small and the graph structure is important, and noisy or missing neighborhood information will hurt model performance. Based on this intuition, we build SCR as:

$$\delta_{MLP} = z_{GNN} - z_{MLP}, \quad \delta_{LP} = z_{GNN} - z_{LP} \tag{5a}$$

$$FCR(\mathcal{G}) = \begin{cases} \frac{\delta_{LP}}{\delta_{MLP} + \delta_{LP}} \times 100\% & z_{MLP} \leq z_{GNN} \\ 1 + \frac{|\delta_{MLP}|}{|\delta_{MLP}| + \delta_{LP}} \times 100\% & z_{MLP} > z_{GNN} \end{cases} \tag{5b}$$

**Interpreting FCR values.** For a particular graph $\mathcal{G}$, if $0\% \leq FCR(\mathcal{G}) < 50\%$, it means $z_{GNN} > z_{LP} > z_{MLP}$, and the neighborhood information in $\mathcal{G}$ is mainly responsible for the GNN achieving good performance. If $50\% \leq FCR(\mathcal{G}) < 100\%$, then $z_{GNN} > z_{MLP} > z_{LP}$, and the node features contribute more to the GNN's performance. If $FCR(\mathcal{G}) \geq 100\%$, then $z_{MLP} > z_{GNN} >$

---

[5]We ignore the node features and use the label logits as explained in (Huang et al., 2020a).

$z_{LP}$, and the node aggregation in GNNs can actually lead to reduced performance compared to pointwise models. This case usually happens for some disassortative graphs, where the majority of neighborhoods hold labels different from that of the center nodes (e.g., as observed by (Liu et al., 2020a)).

**Integrate FCR as a tool to design teacher and student models.** For some graph datasets and models, the SCS generalization can be challenging without neighborhood information (i.e., $z_{GNN} > z_{LP} > z_{MLP}$). We hence consider FCR as a principled "screening process" to select model architectures for both teacher and student that own the best inductive bias for SCS generalization.

To achieve this, during the computation of FCR, we perform exhaustive grid search of the architectures (residual connection types, normalization, hidden layers, etc.) for the MLP, LP, and GNN modules, and pick the best-performing variant. Detailed definition of the search space can be found in Appendix B. We treat this grid search procedure as a special case of architecture selection and hyperparameter optimization for Cold Brew. We observe that FCR is able to identify the GNN and MLP architectures that are particularly friendly for SCS generalization, improving our method design.

In experiments, we observe that different model configurations are favored by different datasets, and we use the found optimal teacher GNN and student MLP architectures to perform Cold Brew. More detailed discussions are presented in section 5.3.

## 5 Experiments and Discussion

In this section, we first evaluate FCR by training GNNs on several commonly used graph datasets and observing how well they generalize to tail and cold-start nodes. We also compare it to the graph homophiliy metric $\beta$ proposed in (Pei et al., 2020). Next, we apply Cold Brew to these datasets and compare its generalization ability to baseline graph-based and pointwise MLP models on these datasets. We also show results on proprietary industry datasets.

### 5.1 Datasets and Splits

We perform experiments on five open-source datasets and four proprietary datasets. The proprietary e-commerce datasets, "E-comm 1/2/3/4", refer to graphs subsampled from anonymized logs of an e-commerce store. They are sampled so as to not reflect the actual raw traffic distributions, and results are provided with respect to a baseline model for these datasets. The different number suffixes refer to different product subsets, and the labels indicate product categories that we wish to predict. Node features are text embeddings obtained from a fine-tuned BERT model. We show FCR values for the public datasets. The statistics of the datasets are summarized in Table 1.

We create training and test splits of the data in order to specifically study the generalization ability of Cold Brew to tail and cold-start nodes. In the following tables, the *head* data corresponds to the top 10% highest-degree nodes in the graph and the subgraph that they induce. We take the data that corresponds to the bottom 10% of the degree distribution, and artifically remove all the edges emanating from these nodes. We then refer to this set of nodes as the *isolation* data. The *tail* data corresponds to the 10% nodes in the remaining graph with lowest (non-zero) degree and the subgraph that they induce. All the zero-degree nodes are in the *isolation* data. The *Overall* data refers to the training/test splits without distinguishing head/tail/isolation.

| | Cora | Citeseer | Pubmed | Arxiv | Chameleon | E-comm1 | E-comm2 | E-comm3 | E-comm4 |
|---|---|---|---|---|---|---|---|---|---|
| Num. of Nodes | 2708 | 3327 | 19717 | 169343 | 2277 | 4918 | 29352 | 319482 | 793194 |
| Num. of Edges | 13264 | 12431 | 108365 | 2315598 | 65019 | 104753 | 1415646 | 8689910 | 22368070 |
| Max Degree | 169 | 100 | 172 | 13161 | 733 | 277 | 1721 | 4925 | 12452 |
| Mean Degree | 4.90 | 3.74 | 5.50 | 13.67 | 28.55 | 21.30 | 48.23 | 27.20 | 28.19 |
| Median Degree | 4 | 3 | 3 | 6 | 13 | 10 | 21 | 15 | 14 |
| Isolated Nodes % | 3% | 3% | 3% | 3% | 3% | 6% | 5% | 5% | 6% |

Table 1: The statistics of datasets selected for evaluation.

### 5.2 FCR Evaluation

In Table 2, the top part presents the FCR results together with the homophily metric $\beta$ from (Pei et al., 2020) (Equation 6). The bottom part shows the prediction accuracies for the head and the tail nodes.

As can be seen from the table, FCR differs among datasets and is negatively correlated with the homophily metric (with Pearson correlation coefficient -0.76). The high absolute correlation value and its negative sign indicate that the more similar the nodes are to their neighborhoods, the more difficult it is to generalize with MLP based models. FCR is thus an indicator of the tail generalization difficulty. Evaluations on more datasets (including the datasets where FCR > 100%) are presented in Appendix C.

$$\beta(\mathcal{G}) = \frac{1}{|\mathcal{V}|} \sum_{v \in \mathcal{V}} \frac{\text{the number of } v\text{'s direct neighbors that have the same labels as } v}{\text{the number of } v\text{'s directly connected neighbors}} \times 100\% \qquad (6)$$

|  | Cora | Citeseer | Pubmed | Arxiv | Chameleon |
|---|---|---|---|---|---|
| GNN | 86.96 | 72.44 | 75.96 | 71.54 | 68.51 |
| MLP | 69.02 | 56.59 | 73.51 | 54.89 | 58.65 |
| Label Propagation | 78.18 | 45.00 | 67.8 | 68.26 | 41.01 |
| FCR % | 32.86 % | 63.39 % | 76.91% | 16.45% | 73.61% |
| $\beta(\mathcal{G})$ % | 83% | 71% | 79% | 68% | 25% |
| $head - tail(GNN)$ | 4.44 | 23.98 | 11.71 | 5.9 | 0.24 |
| $head - isolation(GNN)$ | 31.01 | 33.09 | 15.21 | 28.81 | 1.55 |

Table 2: Top part: FCR and its components. The $\beta$ metric is added as a reference. Bottom part: the performance difference of GNN on the head/tail and head/isolation splits. Here, the "tail/isolation" means the 10% least connected, and isolated nodes in the graph.

## 5.3 EXPERIMENTAL RESULTS ON TAIL GENERALIZATION WITH COLD BREW

In Table 3, we present the performance of Cold Brew together with baselines on the tail and the isolation splits, across several different datasets. All the models in the table are evaluated on the training data, and are evaluated on the tail or isolation splits discussed in section 5.1. In Table 3 *GCN* refers to the the best configuration found through FCR-guided grid search (check Appendix B for details), without Structural Embedding. Correspondingly, *GCN + SE* refers to the best FCR-guided configuration with Structural Embedding, which is the default teacher model of Cold Brew. *GraphSAGE* refers to (Hamilton et al., 2017), *Simple MLP* refers to a simple node-wise MLP that has two hidden layers with 128 hidden dimensions, and *GraphMLP* refers to (Hu et al., 2021). The results for the e-commerce datasets are presented as relative improvements to the baseline (each value is the difference with respect to the value of the *GCN 2 layers* on same dataset of the same split). We do not disclose the absolute numbers due to proprietary reasons.

As shown in Table 3, Cold Brew's student MLP improves accuracy on isolated nodes by up to +11% on the e-commerce datasets and +2.4% on the open-source datasets. Cold Brew's student model handles isolated nodes better, and the teacher GNN also achieves better performance on the tail split compared to all other models. Especially when compared with GraphMLP, Cold Brew's student MLP consistently performs better. This can be explained from their different mechanisms: GraphMLP encodes graph knowledge implicitly in the learned weights, while Cold Brew explicitly attends to neighborhoods even when they are absent. More detailed comparisons can be found in Appendix C.

| Splits | Metrics/Models | | Open-Source Datasets | | | | | Proprietary Datasets | | | |
|---|---|---|---|---|---|---|---|---|---|---|---|
| | | | Cora | Citeseer | Pubmed | Arxiv | Chameleon | E-comm1 | E-comm2 | E-comm3 | E-comm4 |
| Isolation | GNNs | GCN 2 layers | 58.02 | 47.09 | 71.50 | 44.51 | 57.28 | – | – | – | – |
| | | GraphSAGE | 66.02 | 51.46 | 69.87 | 47.32 | 59.83 | +3.89 | +4.81 | +5.24 | +0.52 |
| | MLPs | Simple MLP | 68.40 | **53.26** | 65.84 | 51.03 | 60.76 | +5.89 | +9.85 | +5.83 | +6.42 |
| | | GraphMLP | 65.00 | 52.82 | 71.22 | 51.10 | **63.54** | +6.27 | +9.46 | **+5.99** | +7.37 |
| | Cold Brew | GCN + SE 2 layers | 58.37 | 47.78 | **73.85** | 45.20 | 60.13 | +0.27 | +0.76 | -0.50 | +1.22 |
| | | Student MLP | **69.62** | 53.17 | 72.33 | **52.36** | 62.28 | **+7.56** | **+11.09** | +5.64 | **+9.05** |
| Tail | GNNs | GCN 2 layers | 84.54 | **56.51** | 74.95 | 67.74 | 58.33 | – | – | – | – |
| | | GraphSAGE | 82.82 | 52.77 | 73.07 | 63.23 | 61.26 | -3.82 | -3.07 | -2.87 | -6.42 |
| | MLPs | Simple MLP | 70.76 | 54.85 | 67.21 | 52.14 | 50.12 | -0.37 | +1.74 | -0.13 | -0.45 |
| | | GraphMLP | 70.09 | 55.56 | 71.45 | 52.40 | 52.84 | -0.33 | +1.64 | **+1.27** | +0.80 |
| | Cold Brew | GCN + SE 2 layers | **84.66** | 56.32 | **75.33** | **68.11** | **60.80** | **+0.85** | +0.44 | -0.60 | +1.10 |
| | | Student MLP | 71.80 | 54.88 | 72.54 | 53.24 | 51.36 | +0.32 | **+3.09** | -0.18 | **+2.09** |

Table 3: The performance comparisons on the isolation and tail splits of different datasets. The full comparisons on head/tail/isolation/overall data are in the Appendix C. GCN+SE 2 layers is Cold Brew's teacher model. Cold Brew outperforms GNN and other MLP baselines, and achieves the best performance on the isolation splits as well as some tail splits.

| Splits | Models | Datasets | | | |
|---|---|---|---|---|---|
| | | Cora | Citeseer | Pubmed | E-comm1 |
| Isolation | GCN 2 layers | 34.10 | 50.41 | 51.52 | – |
| | TailGCN | 36.13 | 51.48 | 51.19 | +2.18 |
| | Meta-Tail2Vec | 36.92 | 50.90 | 51.62 | +2.34 |
| | Cold Brew's MLP | **44.59** | **55.14** | **54.82** | **+5.39** |

| Splits | Models | Datasets | | | |
|---|---|---|---|---|---|
| | | Cora | Citeseer | Pubmed | E-comm1 |
| Isolation | GCN 2 layers | 58.02 | 47.09 | 71.50 | – |
| | TailGCN | 62.04 | 51.87 | 72.10 | +3.14 |
| | Meta-Tail2Vec | 61.16 | 50.46 | 71.80 | +2.80 |
| | Cold Brew's MLP | **69.62** | **53.17** | **72.33** | **+7.56** |

Table 4: Link prediction Mean Reciprocal Ranks (MRR) on the isolation data. Note that Cold Brew outperforms baselines specifically built for generalizing to the tail.

Table 5: Node classification accuracies with other baselines specifically created to generalize to the tail. Cold Brew outperforms these methods when edge data is absent in the graph.

We also evaluated the link prediction performance by replacing the node classification loss with the link prediction loss. On the manually created isolation split, the model is asked to recover the ground truth edges which are manually removed. The results are shown in Table 4. The baseline models shown in table are TailGCN (Vetter et al., 1991) and Meta-Tail2Vec (Liu et al., 2020b). A comparison over these models on the node classification on the isolation split is provided in Table 5. As observed from the table 4 and 5, Cold Brew outperformed TailGCN and Meta-Tail2Vec on the isolation split, since both TailGCN and Meta-Tail2Vec explicitly are not zero-shot methods and require explicit neighborhood nodes, hence their performance degrades when the neighborhood is empty and padded by zero vectors.

The full performance on other splits are listed in Table 10 in the appendix as a reference. The results across all splits in Table 10 provide evidence for a few phenomena, for example, the high FCR means that the graph structure does not add too much information for the task at hand, and that GNN type models tend to perform better on the head while MLP type models tend to perform better on the tail/isolation splits. On the other hand, the proposed Structural Embeddings imply a potential to alleviate the over-smoothness (Oono & Suzuki, 2020; Li et al., 2018; NT & Maehara, 2019) and bottleneck (Alon & Yahav, 2020) issues observed in deep GCN models. As shown in table Table 6, Cold Brew's GCN (GCN + SE) significantly outperformed the traditional GCN on 64 layers: the former has 34% test accuracy higher on Cora, 23% higher on Citeseer, and similar on others.

Finally, the improvement over isolation and tail splits (especially the isolation split) comes with a cost: we observed a performance drop for the student MLP model on the head and several other datasets' tail splits, compared with the naive GCN model. However, Cold Brew specifically targets the challenging strict cold start issues, as a new compelling alternative for in these cases. Meanwhile in the non-cold-start cases, the traditional GCN models can still be used to obtain good performance. Note that even on the head splits, the proposed GNN teacher model of Cold Brew still outperformed traditional GNN models. We hence consider as promising future work to adaptively switch between using Cold Brew teacher and student models, based on the current node connectivity degree.

| Splits | Metrics/Models | Open-Source Datasets | | | | | Proprietary Datasets | | | |
|---|---|---|---|---|---|---|---|---|---|---|
| | | Cora | Citeseer | Pubmed | Arxiv | Chameleon | E-comm1 | E-comm2 | E-comm3 | E-comm4 |
| Overall | GCN 64 layers | 40.04 | 23.66 | 75.65 | 65.53 | 58.14 | -5.49 | -6.59 | -6.13 | -3.57 |
| | GCN + SE 64 layers | **74.23** | **46.80** | **78.12** | **69.28** | **59.88** | **-1.71** | **-2.92** | **-3.29** | **-0.06** |
| Head | GCN 64 layers | 46.46 | 49.84 | 85.89 | 67.53 | 67.16 | -5.60 | -6.24 | -6.05 | -3.16 |
| | GCN + SE 64 layers | **87.38** | **71.18** | **86.81** | **71.35** | **69.63** | **-2.17** | **-2.79** | **-0.35** | |
| Tail | GCN 64 layers | 45.14 | 24.42 | 71.89 | 63.91 | 56.48 | -3.85 | -3.62 | -3.84 | **-1.14** |
| | GCN + SE 64 layers | **79.56** | **36.52** | **74.88** | **65.19** | **61.73** | **-2.42** | **-2.52** | **-3.68** | -1.23 |
| Isolation | GCN 64 layers | 39.97 | 22.12 | 68.57 | 40.03 | 57.60 | -4.66 | -4.63 | -4.93 | **-1.89** |
| | GCN + SE 64 layers | **40.33** | **24.53** | **71.22** | **41.18** | **60.13** | **-3.08** | **-3.02** | **-4.00** | -2.32 |

Table 6: The comparisons of Cold Brew's GCN and the traditional GCN for deep layers. When the number of layers is large, Cold Brew's GCN retains good performance while the traditional GCN without SE suffers from the "over-smoothess" and degrades. Even with shallow layers, Cold Brew's GCN is better than traditional GCN.

## 6 CONCLUSION

In this paper, we studied the problem of generalizing GNNs to the tail and strict cold start nodes, whose neighborhood information is either sparse/noisy or completely missing. We proposed a teacher-student knowledge distillation procedure to better generalize to the isolated nodes. We added an independent set of structural embeddings in GNN layers to alleviate node over-smoothness, and also proposed a virtual neighbor discovery step for the student model to attend to latent neighborhoods. We additionally present the FCR metric to quantify the difficulty of truly inductive representation learning and to optimize our model architecture design. Experiments demonstrated the consistently superior performance of our proposed framework on both public benchmarks and proprietary datasets.

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

## A    MORE ILLUSTRATIONS

The more detailed inference procedures for GNN and Cold Brew are illustrated in Figure 3.

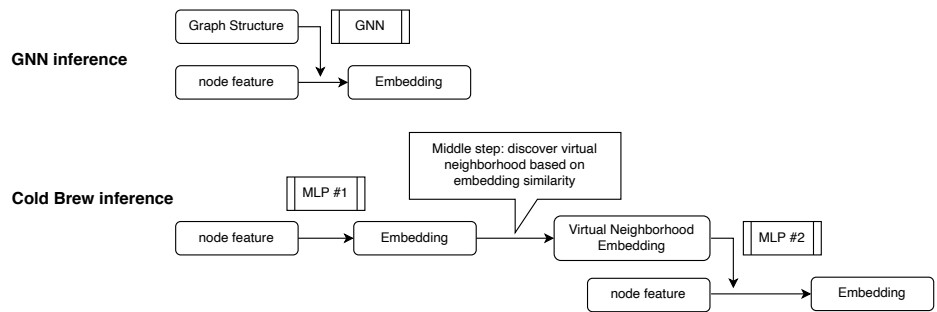

Figure 3: Inference procedure illustration for GNN and Cold Brew.

## B    SEARCH SPACE DETAILS

In computing FCR, we include a search space of model hyperparameters for GNN, MLP, and LP in order to find the best suitable configurations for distillation.

For the GNN model, we take GCN as a backbone and performed grid search over the number of hidden layers, whether it has the structural embedding, the type of residual connection, and the type of normalization. For the number of hidden layers, we considered 2, 4, 8, 16, 32, and 64. For the types of residual connections, we include: (1) connection to the last layer (Li et al., 2019a; 2018), (2) initial connection to the initial layer (Chen et al., 2020a; Klicpera et al., 2018; Zhang et al., 2020), (3) dense connection to all preceding layers (Li et al., 2019a; 2018; 2020; Luan et al., 2019), and (4) jumping connection combining all the preceding layers only at the final graph convolutional layer (Xu et al., 2018b; Liu et al., 2020b). For the types of normalizations, we grid search over batch normalization (BatchNorm) (Ioffe & Szegedy, 2015), pair normalization (PairNorm) (Zhao & Akoglu, 2019), node normalization (NodeNorm) (Zhou et al., 2020b), mean normalization (MeanNorm) (Yang et al., 2020), and differentiable group normalization (GroupNorm) (Zhou et al., 2020a). For types of graph dropout methods, we include Dropout (Srivastava et al., 2014), DropEdge (Rong et al., 2020), DropNode (Huang et al., 2020c), and LADIES (Zou et al., 2019).

For the architecture design for Cold Brew's MLP, we conducted hyperparameter search over the number of hidden layers, the existence of residual connection, the hidden dimensions, and the optimizers. The number of hidden layers is searched over 2, 8, 16, and 32. The number of hidden dimensions is searched over 128 and 256. The optimizer is searched over (Adam(lr=0.001) Adam(lr=0.005), Adam(lr=0.02), SGD(lr=0.005))

For Label Propagation, we conducted hyperparameter search over the number of propagations, the propagation matrix type, and the mixing coefficient $\alpha$ (Huang et al., 2020a). The number of propagations is searched over 10, 20, 50, 100, and 200. The propagation matrix type is searched over adjacency matrix and normalized Laplacian matrix. The mixing coefficient $\alpha$ is searched over 0.01, 0.1, 0.5, 0.9, and 0.99.

The best GCN, MLP, and LP configurations are reported in Tables 7, 8, and 9, respectively.

## C    THE PERFORMANCE ON ALL SPLITS OF THE DATA

The performance evaluations on all splits are listed in Table 10. The FCR evaluation on more datasets are presented in Figure 11. We hypothesize that a high FCR means that the graph does not add

| Dataset | Best GCN | | | |
|---|---|---|---|---|
| | num layers | whether has SE | residual type | normalization type |
| Cora | 2 layer | has structural embedding | no residual | PairNorm |
| Citeseer | 2 layer | has structural embedding | no residual | PairNorm |
| Pubmed | 16 layer | has structural embedding | initial connection | GroupNorm |
| Arxiv | 4 layer | has structural embedding | initial connection | GroupNorm |
| Chameleon | 2 layer | has structural embedding | initial connection | BatchNorm |

Table 7: Best GCN configurations.

| Dataset | Best MLP | | | |
|---|---|---|---|---|
| | hidden layers | residual connection | hidden dimensions | optimizer |
| Cora | 2 layer | no residual | 128 | Adam(lr=0.001) |
| Citeseer | 4 layer | no residual | 128 | Adam(lr=0.001) |
| Pubmed | 2 layer | no residual | 256 | Adam(lr=0.02) |
| Arxiv | 2 layer | no residual | 256 | Adam(lr=0.001) |
| Chameleon | 2 layer | no residual | 256 | Adam(lr=0.001) |

Table 8: Best MLP configurations.

too much information for the task at hand. We indeed see evidence for this hypothesis in Table 10, where for the Pubmed dataset ($FCR \approx 77\%$), the MLP-type models tend to outperform GNN-type models in all splits On the other hand, regardless of FCR, for almost all datasets, the MLP-type models outperform the GNN-type models on the isolation split, and a few on the tail split, while the GNN-type models are superior in other splits.

# D VISUALIZING THE LEARNED EMBEDDINGS

Figure 4 visualizes the last-layer embeddings of different models after t-SNE dimensionality reduction. In the figure, colors denotes node labels and all nodes are marked as dots, with isolation subset nodes additionally marked with $x$'s and the tail subset additionally marked with triangles. Although the GCN model did a decent job in separating different classes, a significant portion of the tail and isolation nodes fall into wrong class clusters. Cold Brew's MLP is more discriminative in the tail and isolation splits.

| Dataset | Best LP | | |
|---|---|---|---|
| | number of propagations | propagation matrix type | mixing coefficient |
| Cora | 50 | Laplacian matrix | 0.1 |
| Citeseer | 100 | Laplacian matrix | 0.01 |
| Pubmed | 50 | Adjacency matrix | 0.5 |
| Arxiv | 100 | Laplacian matrix | 0.5 |
| Chameleon | 50 | Laplacian matrix | 0.1 |

Table 9: Best Label Propagation configurations.

| Splits | Metrics/Models | | Open-Source Datasets | | | | | Proprietary Datasets | | | |
|---|---|---|---|---|---|---|---|---|---|---|---|
| | | | Cora | Citeseer | Pubmed | Arxiv | Chameleon | E-comm1 | E-comm2 | E-comm3 | E-comm4 |
| Isolation | GNNs | GCN 2 layers | 58.02 | 47.09 | 71.50 | 44.51 | 57.28 | – | – | – | – |
| | | GraphSAGE | 66.02 | 51.46 | 69.87 | 47.32 | 59.83 | +3.89 | +4.81 | +5.24 | +0.52 |
| | MLPs | Simple MLP | 68.40 | **53.26** | 65.84 | 51.03 | 60.76 | +5.89 | +9.85 | +5.83 | +6.42 |
| | | GraphMLP | 65.00 | 52.82 | 71.22 | 51.10 | **63.54** | +6.27 | +9.46 | **+5.99** | +7.37 |
| | Cold Brew | GCN + SE 2 layers | 58.37 | 47.78 | **73.85** | 45.20 | 60.13 | +0.27 | +0.76 | -0.50 | +1.22 |
| | | Student MLP | **69.62** | 53.17 | 72.33 | **52.36** | 62.28 | **+7.56** | **+11.09** | +5.64 | **+9.05** |
| Tail | GNNs | GCN 2 layers | 84.54 | **56.51** | 74.95 | 67.74 | 58.33 | – | – | – | – |
| | | GraphSAGE | 82.82 | 52.77 | 73.07 | 63.23 | 61.26 | -3.82 | -3.07 | -2.87 | -6.42 |
| | MLPs | Simple MLP | 70.76 | 54.85 | 67.21 | 52.14 | 50.12 | -0.37 | +1.74 | -0.13 | -0.45 |
| | | GraphMLP | 70.09 | 55.56 | 71.45 | 52.40 | 52.84 | -0.33 | +1.64 | **+1.27** | +0.80 |
| | Cold Brew | GCN + SE 2 layers | **84.66** | 56.32 | **75.33** | **68.11** | 60.80 | **+0.85** | +0.44 | -0.60 | +1.10 |
| | | Student MLP | 71.80 | 54.88 | 72.54 | 53.24 | 51.36 | +0.32 | **+3.09** | -0.18 | **+2.09** |
| Head | GNNs | GCN 2 layers | 88.68 | 80.37 | 85.79 | 73.35 | 67.49 | – | – | – | – |
| | | GraphSAGE | 87.75 | 74.81 | 86.94 | 70.85 | 62.08 | -4.26 | -4.17 | -3.50 | -7.46 |
| | MLPs | Simple MLP | 74.33 | 72.00 | 89.00 | 56.34 | 60.82 | -16.74 | -18.10 | -16.73 | -16.51 |
| | | GraphMLP | 72.45 | 69.83 | 89.00 | 56.65 | 62.44 | -15.96 | -18.08 | -15.33 | -15.41 |
| | Cold Brew | GCN + SE 2 layers | **89.39** | **80.76** | 87.83 | **74.01** | 70.56 | **+1.11** | **+0.47** | -0.39 | **+1.28** |
| | | Student MLP | 74.53 | 72.33 | **90.33** | 57.41 | 61.28 | -15.28 | -17.42 | -17.02 | -15.41 |
| Overall | GNNs | GCN 2 layers | 84.89 | 70.38 | 78.18 | 71.50 | 59.30 | – | – | – | – |
| | | GraphSAGE | 80.90 | 66.21 | 76.73 | 68.33 | 70.02 | -3.09 | -3.86 | -2.58 | -5.48 |
| | MLPs | Simple MLP | 69.02 | 56.59 | 73.51 | 54.89 | 58.65 | -12.69 | -12.86 | -12.68 | -13.16 |
| | | GraphMLP | 71.87 | 68.22 | 82.03 | 53.81 | 57.67 | -12.26 | -12.01 | -10.80 | -11.41 |
| | Cold Brew | GCN + SE 2 layers | **86.96** | **72.44** | 79.03 | **71.92** | **68.51** | **+0.65** | -0.24 | -0.77 | **+1.43** |
| | | Student MLP | 72.36 | 67.54 | **82.00** | 54.94 | 59.07 | -11.25 | -11.51 | -11.55 | -11.21 |

Table 10: The performance comparisons on all splits of different datasets.

| | Cora | Citeseer | Pubmed | Arxiv | Cham. | Squ. | Actor | Cornell | Texas | Wisconsin |
|---|---|---|---|---|---|---|---|---|---|---|
| GNN | 86.96 | 72.44 | 75.96 | 71.54 | 68.51 | 31.95 | 59.79 | 65.1 | 61.08 | 81.62 |
| MLP | 69.02 | 56.59 | 73.51 | 54.89 | 58.65 | 38.51 | 37.93 | 86.26 | 83.33 | 85.42 |
| Label Propagation | 78.18 | 45.00 | 67.8 | 68.26 | 41.01 | 22.85 | 29.69 | 32.06 | 52.08 | 40.62 |
| FCR % | 32.86 % | 63.39 % | 76.91% | 16.45% | 73.61% | 141.91% | 57.93% | 139.04% | 171.2 % | 108.48 % |
| $\beta(\mathcal{G})$ % | 83% | 71% | 79% | 68% | 25% | 22% | 24% | 11% | 6% | 16% |
| $head - tail(GNN)$ | 4.44 | 23.98 | 11.71 | 5.9 | 0.24 | -6.51 | 2.22 | -4.37 | -11.26 | -33.92 |
| $head - isolation(GNN)$ | 31.01 | 33.09 | 15.21 | 28.81 | 1.55 | -4.85 | 22.61 | -18.68 | -24.62 | -29.23 |

Table 11: Top part: FCR and its components. The $\beta$ metric is added as a reference. Bottom part: the performance difference of GNN on the head/tail and head/isolation splits.

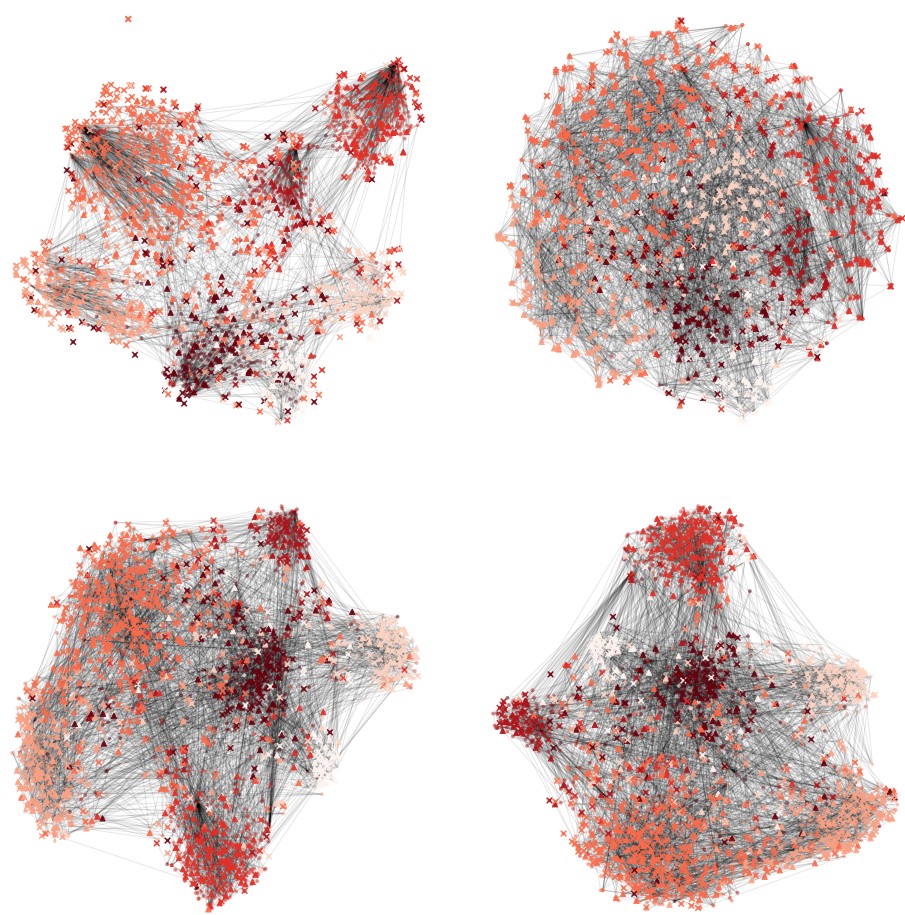

Figure 4: Top two subfigures: the last-layer embeddings of GCN and Simple MLP. Bottom two subfigures: the last-layer embeddings of GraphMLP and Cold Brew's student MLP. All embeddings are projected to 2D with t-SNE. Cold Brew's MLP has the fewest isolated nodes that are misplaced into wrong clusters.

