# OpenReview forum: "Cold Brew: Distilling Graph Node Representations with Incomplete or Missing Neighborhoods"
_ICLR.cc/2022/Conference — ICLR 2022 Poster_

### Official Review · Reviewer_cztX · 2021-10-28

**Correctness:** 3
**Technical Novelty And Significance:** 4
**Empirical Novelty And Significance:** Not applicable
**Recommendation:** 6
**Confidence:** 3

**Main Review:**

Strengths:
1. The paper is well-motivated. The problem studied in this paper is important for the graph domain.
2. The proposed knowledge distillation method sounds interesting and novel.
3. The paper proposes the feature-contribution ratio to guide the selection of model architectures.

Weaknesses:
1. The authors claim that the related works about cold start do not address the case of noisy or missing neighborhoods. However, the paper also focuses on the general cold-start problem and does not address the noisy or missing neighborhoods. I did not find any discussion or designed strategy to handle the noisy or missing neighborhoods explicitly.

2. It is not clear why the structural embedding can encode the label information.

3. The assumption behind the proposed knowledge distillation method is not clear. Can I suppose that the paper has an implicit assumption that is "the nodes with similar features should have similar neighborhoods"? Because the student is to learn a mapping from the node features to $\overline{E}$ that is learned graph structure.

4. The authors claim that the MLP student will behave like the GNN teacher but generalize better to tail and cold-start nodes. But as Table 3 shown, GCN+SE outperforms the student MLP in the tail scenario, and the results need more explanation and discussion.

**Summary Of The Paper:**

The paper proposes Cold Brew to distill the knowledge of a GNN teacher into an MLP student to handle the tail and cold start generalization problem by using the head part of the graph to guide the discovery of the latent neighborhoods of tail and isolation nodes. The paper also proposes a new metric to measure the contribution ratio of node features w.r.t. the adjacency structure. The experiments on several public datasets and a proprietary e-commerce graph show the effectiveness of the proposed method.

**Summary Of The Review:**

The paper provides an interesting and novel solution for the critical cold-start problem. However, several claims are not well-explained or well-supported.

The authors addressed most of my concerns, I would like to update my score.

---

> ### Author Response · Authors · 2021-11-18
> **Authors' response to reviewer cztX**
>
> Dear reviewer cztX:
>
> We appreciate your valuable suggestions to strengthen our paper. We have addressed all raised questions. We hope our explanations depict a clearer picture of our methodologies and better convey the value of this work.
>
> **Q1:** The paper focuses on the general cold-start problem and does not address the noisy or missing neighborhoods.
>
> **Reply:**
> We stress that our paper addresses both the noisy/missing neighborhoods and the cold start problem. One of the assumptions underlying our work (which is confirmed by experiments) is that the node representation for any node can be represented as a (sparse) linear combination of representations of existing nodes in the graph. To this end, our method can be applied to the target node even if the node is missing neighborhood information or has a noisy neighborhood.
>
> In the case of noisy neighborhoods, Cold Brew's student model leverages the “smoothed” knowledge from the teacher GNN so as to mitigate the noise. In the case of missing neighborhoods, Cold Brew's student model directly takes the node features as input. Hence, it addresses the problems of noisy and missing neighborhoods. We offer a novel explanation for this mechanism, the “sparse support recovery”, in the updated PDF. Please kindly find it in the second paragraph of section 3.
>
> We also note that the proposed design is the first to address the truly zero-shot embedding learning problem, where no neighbor is available. Previous studies on the cold start assume that some availability of neighborhood information is sparse, and are hence few-shot embedding learning approaches.
>
> **Q2:** It is not clear why the structural embedding can encode the label information.
>
> **Reply:**
>
> We quote the explanations in our paper and provide a new example to illustrate the reason. As discussed in section 3.1, the structural embedding $E$ can be learned to encode label information through gradient backpropagation. Because it is a node-wise embedding, it can be optimized to bring knowledge from the node label and could reduce the label prediction error. As another reviewer YxwQ mentioned: "E represents the node-wise representation with label information".
>
> Consider a simplified case with only one layer of graph convolution and without nonlinear activation. Denote the input feature as $X$, adjacency matrix as $A$, and the output is $Y=AXW+E$. Then, one solution for this network could be $W=0$, and $Y=0+E$, which means the $E$ is learned to be exactly equal to the node label $Y$.
>
> **Q3:** Can I suppose that the paper has an implicit assumption: "the nodes with similar features should have similar neighborhoods"?
>
>
> **Reply:**
> We would like to note the difference between the virtual neighborhood and the true neighborhood. We assume that isolated nodes should have virtual neighborhoods with similar features. On the other hand, for non-isolated nodes, true neighbors need not to be similar and can be sparse/noisy.
>
> We have another related hypothesis: accurate predictions under noisy neighbors can be achieved via multiple information sources. The proposed model first learns to embed the entire graph by the teacher GNN. During this procedure, the SE makes these embeddings location aware. Finally, the embeddings from the entire graph acts as an overcomplete vector basis in the embedding space, which the student model uses to obtain representations.
>
> **Q4:** The authors claim that the MLP student will behave like the GNN teacher while generalizing better to tail and cold-start nodes, but GCN+SE outperforms the student MLP in the tail for Cora/Citeseer/etc.
>
> **Reply:**
> Thank you for raising this concern, but we would like to clarify that our claim is consistent with our experimental results. In table 3, the e-comm1/2/3/4 datasets are the most desired showcase supporting our claims: they are sampled from the real world user query & product data from an e-commerce platform. In these datasets, we observe that the student MLP performed even better on both the tail and the isolation split.
>
> The open-source citation datasets (Cora/Citeseer/etc.) have all been carefully pre-processed, and all isolated nodes are removed. Therefore, they do not have natural tail/isolation splits. As a result, we artificially created these splits. Even given that, on the isolated splits, the student MLP is still significantly better than the GCN teacher.
>
> The contribution of this paper is the ability to generalize to settings in which the neighborhoods are noisy or missing. Real-world examples are when a new user has only watched a single movie, or a new product has not been purchased yet. In these cases, processing the corresponding nodes with normal graph convolutions will negatively affect the quality of the embeddings for these nodes. These cases have been shown in the results of the e-commerce dataset, where the student achieves the best performance on both tail and isolation nodes.

---

> > ### Author Response · Authors · 2021-11-22
> > **Sincerely expecting further discussions from reviewer cztX**
> >
> >
> > Dear Reviewer cztX:
> >
> > We want to thank for the constructive comments in your review. As a follow-up on our responses, we would like to kindly remind that the discussion period is ending soon. We hope to use this open response period to discuss the paper to solve the concerns and improve the quality of our paper. Have you gotten a chance to read our responses below, which attempt to address all of your concerns?
> >
> > We sincerely hope to have more further discussions with you to see if our response solves the concerns. We would be more than happy to provide more information or clarification, should it be necessary, and hope our paper could receive a positive and fair assessment.
> >
> > Best,
> >
> > Authors of paper 1140

---

> > > ### Comment · Reviewer_cztX · 2021-11-29
> > > **Thanks for the response**
> > >
> > > Thanks for the response. The detailed response addressed most of my concerns. I would like to update my recommendation.

---

> > > > ### Author Response · Authors · 2021-11-29
> > > > **We appreciate your response!**
> > > >
> > > > Dear Reviewer cztX,
> > > >
> > > > We are so glad that our response was able to turn your assessment into a more positive one. Thanks so much!
> > > >
> > > > Regards,

---

> > ### Author Response · Authors · 2021-11-26
> > **Kindly expecting further discussions**
> >
> > Dear reviewer cztX:
> >
> > We kindly noticed that your main concern for this paper is "some claims are not well-explained". In response to this, we have answered them in detail as above, as well as in the updated PDF. As the discussing phase is ending soon, could you please kindly have a look, and let us know if we have addressed your concerns?
> >
> > Looking forward to your reply!
> >
> > Best,
> >
> > Authors of paper 1140

---

### Official Review · Reviewer_svEE · 2021-11-01

**Correctness:** 4
**Technical Novelty And Significance:** 3
**Empirical Novelty And Significance:** 2
**Recommendation:** 6
**Confidence:** 4

**Details Of Ethics Concerns:**

No concerns for the presented work

**Main Review:**

The proposed method to adopt GNN with knowledge distillation to solve SCS problem relies on well-known concepts and previous works, but the learnable Structural Embedding and model selection methodology is novel in contribution and can also have practical adoption for industry applications. The proposed solution is technically sound and is well supported by the extensive evaluation presented. Overall, the paper is easy to follow, and the motivations are also well justified.

The experimental settings and the data preparation are well documented, the authors have performed extensive empirical studies against multiple public datasets and several baseline methods to show their proposed method’s efficacy. The studies consistently show the proposed Cold Brew solution to solve the SCS problem results in significant improvement – especially for the tail & isolation splits.

It should be noted that the focus is primarily on graph-based solutions for solving the SCS problem, which is not the only possible setup. Also, the evaluation and problem formulations are mainly focused on node label prediction (accuracy metric), it would have been interesting to see some metrics like hit-rate (especially for tail & isolated splits of e-commerce datasets) – for ranked recall generation(link-injection) using node representations learned from SCS for a cold-start item which is crucial for generating recommendation.



**Summary Of The Paper:**

The work focuses on a very practical problem of strict cold start(SCS) recommendations which is a highly prevalent and relevant problem. The author’s main contribution to address the SCS is to use GNN with knowledge distillation – this proposed solution does not have to rely exclusively on the node features. The authors also define an FCR (feature contribution ratio) that can help determine the ideal network architecture – FCR can optimize the model selection which significantly affects the quality of overall system performance.


**Summary Of The Review:**

Overall , I think the paper studies an important and interesting problem and presents a good solution which is theoretically sound. It however could try to cover more broader and crucial task especially w.r.t cold-start recommendations .

---

> ### Author Response · Authors · 2021-11-18
> **Authors' response to reviewer svEE**
>
> Dear reviewer svEE:
>
> We sincerely appreciate your acknowledgement of our work and your advice to make our paper stronger. We have improved the narrative and experiments based on your comments.
>
> You mentioned that the focus is primarily on graph-based solutions for solving the SCS problem, which is not the only possible setup. We agree with your point. In fact, the settings could be generalized to not only node classification, but also link prediction in the recommendation system: to ask the model to generate embeddings that lead to high-recall evaluations, and to use these embeddings to build the recommendation system over the tail nodes.
>
> We have conducted further experiments to verify the link prediction performance of Cold Brew and have updated our paper with these new experiment results in table 4 of section 5 and below. To summarize the new results, the student model performed better than the baselines over the link prediction task on the isolation split, since only cold brew is zero-shot method while others are only few-shot methods, and the performance degrades when the neighborhood is empty.
>
> The table below is table 4 in the updated PDF, compared over the isolation splits.
>
> |   | Cora  | Citeseer   |  Pubmed | E-comm1  |
> |---|---|---|---|---|
> | GCN 2 layers  | 34.10  | 50.41  | 51.52  | -  |
> | TailGCN  | 36.13  | 51.48  | 51.19  | +2.18  |
> | Meta-Tail2Vec  | 36.92  | 50.90  | 51.62  |  +2.34  |
> | Cold Brew | 44.59  |  55.14 |  54.82 |  +5.39  |

---

> > ### Author Response · Authors · 2021-11-22
> > **Sincerely expecting further discussions from reviewer svEE**
> >
> >
> > Dear Reviewer svEE:
> >
> >
> > We want to thank you here, again, for the constructive comments and acknowledgment of this paper. We have added the link prediction evaluation results over the e-commerce dataset, and have demonstrated the good performance of the proposed method. We have also improved the presentation of the paper. Could you please kindly check the updated PDF and our response above, to see if your concerns are solved?
> >
> > We would be more than happy to provide more information or clarification, should it be necessary. We hope our paper could receive a positive and fair assessment, to help bring new effective solutions to cold start problems in the graph mining community.
> >
> > Best,
> >
> > Authors of paper 1140

---

> > > ### Comment · Reviewer_svEE · 2021-11-29
> > > **Thank you for the response**
> > >
> > > Thank you for the detailed response and the empirical studies thats been added for the tail-recall generation. This definitely makes the paper more valuable.

---

> > ### Author Response · Authors · 2021-11-27
> > **Kindly expecting further discussions and updated assessment**
> >
> > Dear reviewer svEE:
> >
> > We would like to thank you for the constructive comments in your review. As a follow-up on our responses, we kindly remind you that the discussion period is ending soon. We hope to use this open response period to discuss the paper to solve the concerns and enhance the quality of our work. Have you gotten a chance to read our responses below, which attempt to address all of your issues?
> >
> > We sincerely hope to have further discussions with you to see if our response solves the concerns. We hope our paper could receive a better and fair assessment given the updated results.
> >
> > Best,
> >
> > Authors of paper 1140

---

### Official Review · Reviewer_YxwQ · 2021-11-02

**Correctness:** 3
**Technical Novelty And Significance:** 3
**Empirical Novelty And Significance:** 3
**Recommendation:** 6
**Confidence:** 3

**Main Review:**

Overall, this paper is well-organized. Paying more attention to tail nodes and/or nodes with fewer neighbors is important and has been neglected in a lot of previous studies on GNN. It is also of practical value, e.g., cold start in the recommendation system. Using knowledge distill to learn a transformation from both structures and attributes to attribute is an interesting idea to solve this problem. Experimental results on several graphs demonstrate the effectiveness of the proposed method.

My major concerns are as follows:
- There are some previous studies on tail node representation learning, e.g. [1] and [2]. I suggest that the authors discuss these studies and compare the experimental result with these methods.
- What's the reason for minimizing the structural embedding E in the loss function (Eq (3))? Since E represents the node-wise representation with label information, it is not intuitively clear why this embedding should be as small as possible.
- The experimental studies only validate the performance using GCN. I wonder if the performance will be influenced by different GNN. For example, GraphSage uses a sampling strategy to aggregate embedding, and intuitively this may mitigate the impact of imbalanced distribution or noisy structural information.
- Since label information has been incorporated into the embedding (with the structural embedding E), the ratio of labeled nodes may influence the performance. Could you give some empirical or theoretical analysis of this possible relationship between the performance and ratio of labels?

I also have some minor comments:
- Title: The title contains the terms incomplete or missing, but the main content discusses a more general case including tail nodes, isolated nodes, and nodes with (maybe) incomplete or missing neighborhoods. Please make the title and content consistent.
- Notation z has been used as some element under Eq (4) and performance of models in Eq (5).
- Typo on Page 4: this *motivate* us to strengthen..
- The information in Figure 1 top is clear but the way to show the graph (the grey nodes) on such a coordinate is misleading.

[1] Tail-GNN: Tail-Node Graph Neural Networks, KDD 2021
[2] Towards Locality-Aware Meta-Learning of Tail Node Embeddings on Networks, CIKM 2020

**Summary Of The Paper:**

Many real-world graphs have power-law distributions of node degrees and learning the representations of nodes with few or even no connections may only depend on their attributes. This paper studies the problem of learning good representations of such nodes using inductive GNNs. It proposes a new method to generalize GNNs better for tail nodes compared to pointwise and graph-based
models using a distillation approach. A metric, feature contribution ratio, has been proposed in quantifying the contribution of nodes' features in predicting labels. Experiments on several graph datasets demonstrate the effectiveness of the proposed method especially in learning better representations of the tail and isolated nodes.

**Summary Of The Review:**

This paper studies an interesting and practical problem of learning better representations for tail and isolated nodes. Making use of a distilling approach, the combined information of both structures and attributes can be learned to generalize in nodes with less or even no structural information. Some weakness of the paper includes missing baselines, comparison to other GNNs, and deeper investigation of the label information.
-------------------------------
I appreciate the responses from the authors that addressed most of my concerns. I updated my rating.

---

> ### Author Response · Authors · 2021-11-18
> **Authors' response to reviewer YxwQ (part 2)**
>
> Dear reviewer YxwQ:
>
> Due to length limitation, we continue our response session here.
>
>
> **Q3:** The experimental studies only validate the performance using GCN. I wonder if the performance will be improved by different GNN such as GraphSage.
>
> **Reply:**
>
> Thank you for pointing out this possibility. Using GraphSage as an additional choice of teacher model is a good idea and has the potential to further improve the performance for certain datasets.
>
> Note that the main contribution of the paper is the student-teacher framework to generalize GNNs to tail and isolated nodes, and the specific architecture of the teacher is not as important. A better teacher could indeed help train better student models.
>
> Our teacher model is not a single GCN model. Rather, it is the optimal model over 12 different variant models (selected under the principled FCR metric in a dataset-specific way): PairNorm, NodeNorm, MeanNorm, GroupNorm, Residual, Initial, Jumping, Dense, Dropout, DropEdge, DropNode, and LADIES. The detailed explanations of these 12 baseline methods as well as the performance of the rest of methods in Appendix A.
>
> Table 3 offers a comparison between GraphSage and the two-layer GCN (the best candidate among the above 12 methods). For some datasets, GraphSage outperformed the GCNs, and in other datasets the GCNs performed better. We agree that GraphSage could be added as the 13th teacher model candidate, and we also agree that the performance of the student model will be better if the teacher model is better. In fact, the latter is one of our assumptions, as we are already choosing the best across the 12 as the backbone of the teacher model of Cold Brew. We plan to further include it into the main text.
>
>
>
>
> **Q4:**
>
> the ratio of labeled nodes may influence the performance. Could you give some empirical or theoretical analysis of this possible relationship between the performance and ratio of labels?
>
> **Reply:**
>
> We offer empirical analysis to this question. We run new experiments that change the labeled node ratio, and provide the results for the Cora dataset. As shown in the table below, Cold Brew’s teacher still performs the best for different label ratios.
>
> Isolation split:
>
> | label ratio   |  10%  | 20%  | 40%  | 80%        |
> | :------------- | :----------: | -----------: | | |
> | Cold Brew's Student | 68.36 | 69.62 | 70.14 | 70.80 |
> | Cold Brew's Teacher | 55.73 | 58.67 | 58.90 |  59.21  |
> | GCN 2 layers | 54.26 | 57.84 | 58.44 | 58.54 |
> | GraphMLP | 63.13  | 65.12 | 66.20 | 66.10 |
>
>
>
>
> **Q5:** Other comments on minor improvements/typos:
>
> **Reply:**
> We appreciate your valuable comments to improve our paper. We have made changes accordingly.

---

> > ### Author Response · Authors · 2021-11-26
> > **Kindly looking forward to further discussions**
> >
> > Dear reviewer YxwQ:
> >
> > We noted that the main weakness of the paper you mentioned in your review is: "missing baselines comparison to other GNNs, and deeper investigation of the label information". In response to these points, we have executed new experiments and added discussions, in both the updated PDF and our responses above. Could you please kindly have a check?
> >
> > The table below is the comparison with two baselines: TailGNN and MetaTail2Vec, copy-pasted from table 4 in the updated PDF. It compared over the isolation splits and showed the superior inductive performance of cold brew.
> >
> >
> > |   | Cora  | Citeseer   |  Pubmed | E-comm1  |
> > |--|--|--|--|--|
> > | GCN 2 layers  | 34.10  | 50.41  | 51.52  | -  |
> > | TailGCN  | 36.13  | 51.48  | 51.19  | +2.18  |
> > | Meta-Tail2Vec  | 36.92  | 50.90  | 51.62  |  +2.34  |
> > | Cold Brew | **44.59**  |  **55.14** |  **54.82** |  **+5.39**  |
> >
> >
> > We also note that the discussion phase is ending soon. We would sincerely like to know if there is any updated evaluation/comment, based on these new results?
> >
> > Looking forward to your reply!
> >
> > Best,
> >
> > Authors of paper 1140

---

> ### Author Response · Authors · 2021-11-18
> **Authors' response to reviewer YxwQ (Part 1)**
>
>
> Dear reviewer YxwQ:
>
>
> Thank you for your suggestions to improve the paper. We have attached detailed responses to each question and added new experiments, and we hope our responses help to better capture the value of this paper.
>
> **Q1:** Discussing and comparing the previous studies on tail node representation learning is recommended.
>
> **Reply:** We have added both methodology and experimental comparison in table 4&5 of section 5 of the updated version. TailGNN, meta-tail2vec, and Cold Brew have different methodologies and can all handle the tail nodes. However, as shown in the results, only Cold Brew can handle completely isolated node splits (zero edge connections), due to each model’s assumptions and design.
>
>
> The TailGNN learns a flexible neighborhood translation function between the center node and the neighbor nodes, which is assumed to be transferable across different parts of the same graph. They further use this assumption to tie the target nodes on the tail part from the head part. The TailGNN did not inject any estimation/manipulation of the adjacency structure (they assume the adjacency structure to be static and unchangeable), hence, on the isolated part of the graph, such methodology immediately fails due to the same reason that GNNs fail: the empty neighborhood lead to the inability to conduct the proposed neighborhood translation.
>
> The meta-tail2vec paper formulated the goal of learning tail node embeddings as a few-shot regression problem: given the few links on each tail node, the meta learning model is able to efficiently infer high quality node embeddings on the tail part. It is worth to note that they do apply structural (edge connectivity) manipulations, but the only manipulation of the edge connectivity is the link dropouts applied to the head nodes, which aims to approximate the low edge connectivity for the tail nodes. On the tail nodes, they didn't take any approach to encourage the edge connectivity to grow or to change. In other words, the proposed method is a few-shot meta learning method but not zero shot: it still assumes that the edge connectivity is non-empty. In comparison, the cold brew can be understood as a zero-shot embedding learning methodology, which embraces broader applicability under the long tailed real world data, which has no shortage of isolated cold start nodes.
>
>
>
>
>
> **Q2:** What's the reason for minimizing the structural embedding E in the loss function?
>
> Reply:
> We offer both theoretical and empirical answers to this question. As discussed in the last paragraph of section 3 (the "Additional remarks on the motivations" paragraph) of the originally submitted version, or in the subsection 3.3 of the updated version, the role of the structural embedding is to encode the residual error, which is bounded by the difference of the node features.
>
>
> Recalling those discussions here, Theorem 1 in [a] describes that, at some specific layer, given the node embeddings $X$ generated from the feature, the errors of the label predictions are upper bounded by the difference between the ground truth and the $AXW$, where $A$ and $W$ are the adjacency matrix and weights, respectively. Given this theorem, the role of the structural embedding $E$ is to give the model more flexibility to mitigate this residual error. In other words, the structural embedding is not supposed to encode as much information as possible (otherwise, it is prone to overfit to the training set), but instead to enable richer expressiveness of the model. During training, the structural embedding for all nodes can receive gradients and are trained and smoothed to mitigate residual error. During inference, the nodes in the testing set first obtain an initial embedding from $AXW$. After that, their learned structural embedding can further reduce their error.
>
> Empirically, we applied different levels of regularization to the weights $E$. We observed that an appropriate regularizer yielded the best results, and a regularization of both 0 and $\infty$ downgraded performance slightly.
>
> [a] Hongwei Wang and Jure Leskovec. Unifying graph convolutional neural networks and label propagation. arXiv preprint arXiv:2002.06755, 2020.

---

> > ### Author Response · Authors · 2021-11-22
> > **Follow-up and a kind reminder**
> >
> >
> >
> > Dear Reviewer YxwQ:
> >
> > We want to thank for the constructive comments in your review. As a follow-up on our responses, we would like to kindly remind that the discussion period is ending soon. We hope to use this open response period to discuss the paper to solve the concerns and improve the quality of our paper. Have you gotten a chance to read our responses below, which attempt to address all of your concerns?
> >
> > In response to your suggestion on discussing and comparing previous studies on tail node representation,  we have added TailGNN and Meta-Tail2Vec in both our updated PDF and the response. We have answered your questions regarding minimizing the structural embedding E and using GraphSAGE as teacher model. For the influence of the ratio of labeled nodes, we have added a new set of experiments, and reported it in our response Q4.
> >
> > We have also improved the overall presentation of the paper, and have addressed all typos you mentioned. Please kindly have a check on the updated PDF.
> >
> > We sincerely hope to have more further discussions with you to see if our response solves the concerns. We would be more than happy to provide more information or clarification, should it be necessary, and hope our paper could receive a positive and fair assessment.
> >
> > Best,
> >
> > Authors of paper 1140

---

> > > ### Comment · Reviewer_YxwQ · 2021-11-28
> > > **Updated rating**
> > >
> > > Thank you for your detailed responses especially the extra empirical studies. I would like to update my rating.

---

> > > > ### Author Response · Authors · 2021-11-29
> > > > **Thank you**
> > > >
> > > > Dear reviewer YxwQ:
> > > >
> > > > Thank you for your acknowledgement of this work!
> > > >
> > > > Best wishes!
> > > >
> > > > Authors of paper 1140

---

### Official Review · Reviewer_kTEC · 2021-11-03

**Correctness:** 3
**Technical Novelty And Significance:** 2
**Empirical Novelty And Significance:** 2
**Recommendation:** 6
**Confidence:** 4

**Main Review:**

Strength

1 - Propose a knowledge-distillation based technique for learning cold start node representation.

2 - The problem is relatively new and interesting.

3 - Presentation is overall good.

Weakness

1 - The novelty of proposed model is not significant.

2 - Experiments could be improved.

3 - Lacks related work discussion.

Detailed Review

It is interesting to develop new method for GNN to handle nodes without neighbors. The proposed knowledge distillation framework seems reasonable for me. Besides, FCR metric is proposed to measure the importance of node feature and graph structure. The proposed method works well for node classification task in several datasets. Following are some issues.

The novelty of proposed method is not significant as it follows general teacher-student network and combines both neighbor aggregator and structure embedding to learn node embeddings. These parts are borrowed from existing techniques. It would be better to discuss model contribution. In addition, the current manuscript only studies node classification task while it is also possible to study link prediction task over cold-start nodes (e.g., tail nodes). Moreover, there are some existing works studying tail node representation learning that should be discussed or compared, such as:

Towards locality-aware meta-learning of tail node embeddings on networks, CIKM'21

Tail-GNN: Tail-Node Graph Neural Networks, KDD'21

Minor issues exist, such as typo. For example, the first sentence of section 3.2 should be: to integrate the knowledge of GNN teacher.

--
Update after rebuttal: The authors addressed some of my concerns in experiments. The novelty is still incremental for me. I change my score to borderline above.

**Summary Of The Paper:**

This paper proposes Cold Brew, a new method for learning cold start node embeddings in graphs. Cold Brew leverages teacher-student framework (knowledge distillation) to handle nodes without neighbors by transferring knowledge of teacher network (learned from head nodes) to student network (for tail or isolated nodes). Experiments are conducted to show that the proposed method outperforms some baseline methods.

**Summary Of The Review:**

This work studies an interesting problem. The proposed method is reasonable for solving the problem. The novelty is not significant. In addition, experiments could be improved.

---

> ### Author Response · Authors · 2021-11-18
> **Authors' response to reviewer kTEC**
>
> Dear reviewer kTEC:
>
> We appreciate your acknowledgement of the value of our research object and the presentation of this work. We are excited about this work’s novelty, technical soundness, and wide applicability. We have provided detailed answers to your concerns, and we hope our explanations have drawn a clearer picture of the methodology and the theoretical value of this work.
>
> **Q1:** The novelty of the proposed method is not significant as it follows general teacher-student knowledge distillation from existing techniques.
>
> **Reply:**
> Our methodology is not vanilla knowledge distillation (KD), and the KD part is not the most crucial reason that leads to the success of our methodology. The key is the latent neighborhood discovery step, which takes place between the first and second MLPs of Cold Brew student model. We also offer a novel perspective to understand the latent/virtual neighborhood discovery procedure: the sparse support recovery. Details can be found in the 2nd paragraph of section3 in the updated PDF.
>
> Cold Brew is the first model to tackle the zero-shot cold start problem (other cold start works are few-shot, tackling tail nodes and not isolated nodes). With the help of the latent neighborhood discovery step, Cold Brew can infer embeddings for nodes that are  isolated in the graph, making it fundamentally different from graph methods that explicitly rely on message passing.
>
> We believe this work is with novelty and solid technical contribution. As quoted from reviewer cztX: “The paper provides an interesting and novel solution for the critical cold-start problem.” Quoted from reviewer svEE: “Overall , I think the paper studies an important and interesting problem and presents a good solution which is theoretically sound.” Quoted from reviewer YxwQ, “learning …  is an interesting idea to solve this problem.”
>
> **Q2:** The current manuscript only studies the node classification task while it is also possible to study the link prediction task.
>
> **Reply:** We appreciate your advice to strengthen our paper, and we strongly agree with it. As you suggested, under a broader perspective of tail/isolated node embedding learning, Cold Brew naturally applies several tasks that involve node embeddings, and not just node classification.
>
> We have appended the results of the link prediction task in table 4 of section 5 in the new version and below. In a nutshell, the student model of Cold Brew performs the best among all methods across three public datasets and one e-commerce dataset.
> The table below is table 4 in the updated PDF, compared over the isolation splits.
>
> |   | Cora  | Citeseer   |  Pubmed | E-comm1  |
> |--|--|--|--|--|
> | GCN 2 layers  | 34.10  | 50.41  | 51.52  | -  |
> | TailGCN  | 36.13  | 51.48  | 51.19  | +2.18  |
> | Meta-Tail2Vec  | 36.92  | 50.90  | 51.62  |  +2.34  |
> | Cold Brew | 44.59  |  55.14 |  54.82 |  +5.39  |
>
> **Q3:** More methods could be compared/discussed such as the TailGNN and the meta-tail2vec.
>
> **Reply:** We have added both methodology and experimental comparison in table 4&5 of section 5 of the updated version. TailGNN, meta-tail2vec, and Cold Brew have different methodologies and can all handle the tail nodes. However, as shown in the results, only Cold Brew can handle completely isolated node splits (zero edge connections), due to each model’s assumptions and design.
>
> The TailGNN learns a flexible neighborhood translation function between the center node and the neighbor nodes, which is assumed to be transferable across different parts of the same graph. They further use this assumption to tie the target nodes on the tail part from the head part. The TailGNN did not inject any estimation/manipulation of the adjacency structure (they assume the adjacency structure to be static and unchangeable), hence, on the isolated part of the graph, such methodology immediately fails due to the same reason that GNNs fail: the empty neighborhood lead to the inability to conduct the proposed neighborhood translation.
>
> The meta-tail2vec paper formulated the goal of learning tail node embeddings as a few-shot regression problem. It is worth to note that they do apply structural (edge connectivity) manipulations, but the only manipulation of the edge connectivity is the link dropouts applied to the head nodes, which aims to approximate the low edge connectivity for the tail nodes. On the tail nodes, they didn't take any approach to encourage the edge connectivity to grow or to change. In other words, the proposed method is a few-shot meta learning method but not zero shot: it still assumes that the edge connectivity is non-empty. In comparison, the cold brew can be understood as a zero-shot embedding learning methodology, which embraces broader applicability under the long tailed real world data, which has no shortage of isolated cold start nodes.
>
> **Q4:** Minor typos exist.
>
> **Reply**: Thank you for helping us pick them out. We have made another thorough check and have cleared these typos.

---

> > ### Author Response · Authors · 2021-11-22
> > **Sincerely expecting further discussions from reviewer kTEC**
> >
> > Dear Reviewer kTEC:
> >
> > We sincerely appreciate your time for the review and your constructive comments. We hope to use this open response period to discuss the paper, answer additional questions, and ultimately improve the quality of our submission. Have you gotten a chance to read our responses below, which attempt to address all of your concerns?
> >
> > Regarding your concerns for link prediction tasks and comparison with TailGNN/Meta-Tail2vec, we have conducted new experiments, and have updated the results both in the paper and in the response. Regarding your concern for the typos, we have modified them and have improved the overall presentation of the paper.
> >
> > We sincerely hope to have more further discussions with you to see if our response solves the concerns.
> >
> > Best,
> >
> > Authors of paper 1140

---

> > > ### Comment · Reviewer_kTEC · 2021-11-23
> > > **thank you for reminder.**
> > >
> > > see updated review and score.

---

> > > > ### Author Response · Authors · 2021-11-24
> > > > **Thank you**
> > > >
> > > > Dear reviewer kTEC:
> > > >
> > > > Thank you for your updated score and review!
> > > >
> > > >
> > > > Authors of paper1140

---

### Author Response · Authors · 2021-11-29
**Pre-decision: Summary of updates from Authors**

Dear AC panel and reviewers:

We are glad that the merits of our work have been recognized by all reviewers. We have responded to all reviewers point-by-point, and everyone has now acknowledged a positive assessment after reading our response.

We are summarizing the major points about the paper (post-rebuttal) for a quick understanding for all. This paper
studies an "*important and interesting*" problem (quoting reviewers cztX/YxwQ) and "*presents a good solution which is theoretically sound*" (quoting reviewer svEE). The proposed Cold Brew targets at the challenging *zero shot* cold start problem for the first time.

1. We have included TailGNN[1] and Meta-Tail2Vec[2] into the discussions of updated PDF. In summary, TailGNN and Meta-Tail2Vec relied on center to neighbor translation or few-shot meta learning methods that quickly adapt to tail nodes, but still are not zero shot methods as the proposed Cold Brew, and not generalizing better when the neighborhood is missing.

2. We experimented on link prediction tasks, and involved the comparison of TailGNN and Meta-Tail2Vec with Cold Brew on both node classification and link prediction tasks. The experimental results showed that Cold Brew performed the best among these methods and traditional GCN methods.

3. We have addressed the few technical questions posted by the reviewers point by point, and have added these discussions in the updated PDF, including the new explanation of the mechanism of the proposed method under sparse support recovery.

Given the importance of the challenging cold start problem, the technical novelty/soundness, and the experimental superiority of the proposed method, we genuinely hope the discussions above and below could lead to a positive and fair assessment of the proposed method.


Sincerely,

Authors of paper1140


References

[1] Liu Z, Nguyen T K, Fang Y. Tail-GNN: Tail-Node Graph Neural Networks[C]//Proceedings of the 27th ACM SIGKDD Conference on Knowledge Discovery & Data Mining. 2021: 1109-1119.

[2] Liu Z, Zhang W, Fang Y, et al. Towards locality-aware meta-learning of tail node embeddings on networks[C]//Proceedings of the 29th ACM International Conference on Information & Knowledge Management. 2020: 975-984.

---

### Decision · Program_Chairs · 2022-01-20

**Decision:**

Accept (Poster)

**Comment:**

The reviewers agree that the paper studies an important and interesting problem and presents a good solution which is theoretically sound. The paper can be further improved by looking into more applications such as cold-start recommendations.